# A New Way to Model Periodontitis in Laboratory Animals

**DOI:** 10.3390/dj11090219

**Published:** 2023-09-18

**Authors:** Denis Moiseev, Sergey Donskov, Ivan Dubrovin, Mariya Kulyukina, Yuriy Vasil’ev, Beatrice Volel, Shodiya Shadieva, Aleksey Babaev, Juliya Shevelyuk, Anatolij Utyuzh, Ellina Velichko, Sergey Dydykin, Irina Dydykina, Yuri Paramonov, Ekaterina Faustova

**Affiliations:** 1N.I. Pirogov Russian National Research Medical University, Ostrovitianov Str. 1, 117997 Moscow, Russia; moiseeff.den@yandex.ru (D.M.);; 2Tver State Medical University, Sovetskaya Str., 4, 170100 Tver, Russia; donskov_s@mail.ru (S.D.); dubrovin-i@mail.ru (I.D.);; 3I.M. Sechenov First Moscow State Medical University (Sechenov University), Trubetskaya Str., 8, p. 2, 119048 Moscow, Russialinavel83@gmail.com (E.V.);; 4Bukhara State Medical Institute Named after Abu Ali ibn Sino, A.Navai Str. 1, Bukhara 200101, Uzbekistan; 5N.A. Semashko National Research Institute of Public Health, Vorontsovo Pole Str., 12, Building 1, 105064 Moscow, Russia; 6V.A. Nasonova Research Institute of Rheumatology, 34A Kashirskoe Highway, 115522 Moscow, Russia

**Keywords:** animal model, experimental induced periodontitis, histologic analysis, inflammation

## Abstract

The prevalence of periodontal diseases is increasing, tends to increase with age and is considered as one of the main causes of tooth loss. To assess the effectiveness of new methods of treatment of periodontal diseases, studies on laboratory animals can be promising. The aim of the study: to develop a new method of accelerated modeling of experimental periodontitis on laboratory animals. Material and methods. The study was carried out on 22 female rats. A wire ligature was applied to the cervical area of the incisors of the animals in an eight-shaped manner. Plaque obtained from a patient with periodontitis was placed under the wire, and nicotine and ethyl alcohol solutions were injected under the gingival mucosa. A complex index has been proposed to assess inflammation. At the end of the experiment the animals were euthanized, their jaws were dissected into dentoalveolar blocks and further descriptive histologic analysis was performed. Results. On the second day the gingiva of the rats acquired a cyanotic-pink color, on the fourth day the consistency of the gingiva became friable, mobility appeared in the lower incisors. Complex index of inflammation in animals of the main group: before the study—9, on the 7th day—195. Gingival preparations showed signs of exudative inflammation. In alveolar processes—irreversible resorption of bone structures. The difference of indicators in animals before and after the experiment was statistically significant (*p* < 0.05). Conclusion. The new experimental model of periodontitis is reproduced in a short period of time, provides intensive development of inflammation, leads to disruption of the integrity of epithelial and connective tissue attachment, destruction of alveolar bone.

## 1. Introduction

Periodontitis is an infectious disease that leads to destruction of tooth support structures, gradual loss of epithelial, connective tissue attachment and bone tissue of the jaw, appearance of periodontal and bone pockets, characterized by prominent dysbiosis in the oral cavity. Severe periodontitis is the 6th most common (11.2% of the global population) disease worldwide [1]. The results of a recent epidemiologic study conducted in 27 countries indicate that the prevalence of periodontal disease in adults (35–44 years) is about 73%, and among the elderly (65–74 years)—up to 82% and is considered as one of the most frequent causes of tooth loss [2]. In a study published in 2022 involving 431 subjects, with a mean age of 35.4 years, the prevalence of periodontitis among those studied was 85.4% [3]. To date, the role of polymicrobial association in the development of periodontitis remains a priority, with the host inflammatory response playing a decisive role in the development and progression of the disease. Periodontopathogenic microbiota, to which the leading etiological significance is attributed, refers to gram-negative anaerobic bacteria such as, *Porphyromonas gingivalis*, *Aggregatibacter actinomycetemcomitans*, *Tanerella forsythia*, *Treponema denticola*, *Fusobacterium nucleatum* and some others. Their destructive action, caused by unique virulence factors, triggers an inflammatory process in periodontal tissues, which is accompanied by both local reactions and general changes in tissue metabolism with an increase in the level of pro-inflammatory mediators [4].

Periodontitis develops in stages and includes several phases: primary colonization by peculiar bacterial agents, formation and maturation of microbial biofilm, invasion of periodontal tissues by oral microbiota and their metabolites, induction of the host organism, destruction of periodontal tissues with secondary changes in the dentoalveolar complex [1].

Today, a few authors conduct studies that open new pages in understanding the infectious origin of periodontitis and the primary infection in endo-periodontal lesions, offer new methods of diagnostics and treatment of periodontitis [5,6,7]. However, it is difficult to study the efficacy of new treatments and drugs on patients and is associated with several limitations. In this case, it is necessary to carry out investigations on laboratory animals [1,8,9,10,11,12,13].

The complexity and diversity of factors influencing the separate phases of periodontitis pathogenesis, including a combination of polymicrobial synergy and dysbiosis, chronic inflammatory dysregulation, and genetic predisposition factors [4,14,15], have made the search for an ideal animal model of periodontitis difficult and challenging [9]. In addition, the pathogenesis of comorbid conditions aggravated by periodontitis is still poorly understood.

Over the last decades, several fundamentally new models of periodontitis in laboratory animals have been created, which are used to study the pathogenesis of periodontitis and the efficacy of new treatment methods. With all the variety of animal models, the most used is the small rodent model because of the ease of care and handling and the possibility of developing a highly reproducible inflammatory process in the periodontium. There are known methods of obtaining a model of chronic periodontitis in rats by applying acute mechanical trauma to the gingiva [16], by creating an artificial dental plaque using dental cement [17,18] and several others. Among the various methods, the most frequently used is the model of periodontitis induced by ligature, the placement of which around the cervical areas of the animal’s teeth leads to bacterial colonization with the subsequent development of inflammation [1,19,20,21,22]. All these methods have disadvantages: the duration of the experiment, significant traumatization and stress for the animal, and difficulty of reproduction.

Considering the shortcomings of existing methods of modeling experimental periodontitis, to study the pathogenesis of periodontitis and to evaluate the effectiveness of new methods of treatment of inflammatory periodontal diseases, we theoretically substantiated and tested a new experimental model of periodontitis in laboratory rats [23,24].

The aim of the study: to develop a new method of accelerated modeling of experimental periodontitis on laboratory animals.

Null hypothesis: When applying a new method of accelerated modeling of periodontitis on laboratory animals, each one has periodontitis.

## 2. Materials and Methods

Several parameters must be considered to determine the sample size and calculating the number of animals required using “Power and Sample Size Analysis” approaches may result in an unreasonably high number of animals included in the experiment. However, the inclusion of such many animals is neither humane nor economically feasible. In recent years, the European Union has developed approaches that allow us to study pathological processes in animals, maximally limiting the number of animals involved in the experiment (OECD protocols 420, 423, 425) [25,26,27]. Therefore, to determine the sample size, we were guided by recommendations that are the result of many years of work by specialized specialists [28,29,30]. It was decided to limit the number of animal specimens to 22 included in the experiment.

The material for the study was 22 female white rats of the Vistar line, weighing 170–220 g. Experimental studies were conducted in compliance with the International Principles of the European Convention for the Protection of Vertebrate Animals Used for Experiments and Other Scientific Purposes (Strasbourg, France, 1986), in accordance with the General Ethical Principles of Animal Experiments (Russia, 2011), the rules of laboratory practice in the Russian Federation (Order of the Ministry of Health of the Russian Federation NO. 267 of 19 June 2003), in accordance with GOST 33215-2014 “Guidelines for the Keeping and Care of Laboratory Animals. Rules of equipment of premises and organization of procedures” (Interstate Standard, 2016). Primary documentation was maintained in accordance with the principles of Good Laboratory Practice (GOST 33044-2014, Russia).

One of the items of Primary documentation was devoted to the assessment of the animal’s condition during the experiment. Taking into account the trauma in the oral cavity, we defined the points of observation of animal behavior and attempts to remove the ligature from the oral cavity, as well as loss of appetite, forced postures, and vocal accompaniment. Due to the fact that the subjects did not change their eating habits in the postoperative period, did not stay in a forced position, and did not attempt to influence the oral cavity in any way, we concluded that the developed method did not cause painful irritation [31,32,33].

The study was authorized by the local Ethics Committee of FGBOU VO Tver State Medical University of the Russian Ministry of Health (excerpt from the protocol of the meeting No. 5 of 19 June 2020). Rats were kept in the vivarium, under standard conditions (humidity 50–60%, controlled temperature 20–22 °C and 12-h light cycle). All rats were housed in standard cages in groups of two animals, were on a high-carbohydrate diet according to Evdokimov without water restriction. All animals were randomly divided into two groups: main group (11 rats) and control group (11 rats). The experiment included 4 dentoalveolar segments (DAS) of each rat: two incisors each on the upper and lower jaw, a total of 88 DAS (44 in the main group and 44 in the control group). The choice of teeth was related to their accessible location for manipulation. All procedures were performed under combined intramuscular anesthesia using Zolazepam (1 mg/kg) and Xylazine hydrochloride (0.05–0.1 mL/kg). The average duration of the animal’s stay under anesthesia was 1 h [34].

Animals of the main group under anesthesia had a wire ligature with a diameter of 0.3 mm applied in an eight-shaped pattern to the cervical region of the upper and lower incisors. The edges of the wire were twisted and bent, not traumatizing the mucous membrane. Dental plaque from a patient with severe chronic periodontitis, collected by scraping from the crown and cervical areas of the teeth, transported for no more than 2 h in a sterile container, was placed under the wire using a dental spatula. Injection of nicotine solution in physiological solution (nicotine concentration 0.001 mg/L) in the volume of 0.05 mL was carried out under the gingival mucosa on the vestibular side, in the teeth under study, and after 10 min—of 50% ethyl alcohol solution in the volume of 0.05 mL. Injections were performed daily, for 7 days. In animals of the main group, the ligature was checked every day, and if necessary, it was repositioned to keep it during the whole period of the experiment. Animals of the control group under anesthesia were examined daily for 7 days for oral cavity with assessment of dental status. Animals of the control group did not receive any experimental influence on periodontal tissues. The course of the experiment was described in the observation diary, and the condition of the animals was recorded daily [24]. A few dental parameters were evaluated during animal examination: changes in gingival color and consistency, the presence of gingival bleeding when probing, the presence and severity of tooth mobility, the depth of probing of the gingival sulcus and periodontal pocket. Tooth mobility was assessed using Periotest S (Medizintechnik Gulden, Modautal, Germany) and dental forceps. The presence of the sign was evaluated as 1 point, the absence—0 points. In the table of results, the number of animals that received 1 point (presence of the sign) and separately the number of dentoalveolar segments (DAS) that received 1 point were recorded for each indicator. To assess the intensity of inflammation, a new complex index was proposed, which was calculated by adding all values (separately the number of animals and the number of teeth) for the main and control groups and analyzed as a sum.

At the end of the experiment, all animals were euthanized by narcotic drugs overdose in accordance with the Recommendations for the euthanasia of experimental animals (European Commission Expert Group Document, 1997) [35] and in accordance with GOST 33215–2014 “Guidelines for the maintenance and care of laboratory animals. Rules for equipment of premises and organization of procedures” (Interstate Standard, 2016). The animals were euthanized in the following turn: ether sedation, subcutaneous injection of zolazepam and xylazine hydrochloride in doses of 8.0 mg/kg and 4.0 mg/kg, respectively, provision of vascular access (by puncture of the dorsal caudal vein) and bolus injection of lidocaine hydrochloride in a dosage of 100 mg/kg [36]. After three times (at intervals of 5, 10, and 15 min after the bolus) determination of circulatory arrest by the veterinary monitor M6S (TooToo Meditech, Shenzhen, China), we proceeded to autopsy. The time limit from the moment of clinical death to the beginning of histologic material fixation was 2 h.

The upper and lower jaw of each animal was dissected into dentoalveolar blocks; autopsy material was fixed in 10% buffered formalin. After washing it in water, it was dehydrated in 8 portions of isopropyl alcohol and poured into Histomix paraffin medium (Biovitrum, Saint Petersburg, Russia). The following equipment was used for this stage: histological material wiring battery, thermostat TS—1/20 (Smolensk SKTB SPU, Smolensk, Russia), Lamsystems extraction system (Laminar Systems, Russia). Paraffin embedding was performed using the ESD—2800 embedding station (MedTechnikaPoint, Saint Petersburg, Russia). To make slices of the paraffin block with the investigated material enclosed in it, a semi-automatic rotary microtome Hestion ERM 3100 (Hestion Scientific Pty Ltd Melbourne, Australia) was used. Sagittal slices 5 μm thick were obtained and stained with hematoxylin and eosin for further descriptive histologic analysis [37]. Microscopic study of the obtained experimental material was performed using an Olympus CX21 microscope (Olympus Corp., Tokyo, Japan) at low (×10), high (×40) magnifications and in oil immersion (×100). Microphotographs were taken with a MC-10 digital camera (LOMO, Saint Petersburg, Russia), and computer processing of the images was performed with MCview software version: 4.0.1.

During observations, the features of the course of experimental periodontitis were recorded, noting the possibilities of prospective evaluation of treatment and prophylaxis methods on the model.

Statistical processing of data was performed in MS Excel program. The Wilcoxon t-criterion was chosen for statistical processing. Statistical significance was determined at the level of *p* ≤ 0.05.

## 3. Results

### 3.1. Dental Examination

As a result of the study, it was found that the condition of the gingiva of the rats of the main group changed over time (Table 1). Thus, initially the gingiva was pale pink in color (two animals had slight gingival cyanosis and one animal had hyperemia), the gingival sulcus was not probed, gingival bleeding was absent. The lower incisors had insignificant mobility, up to 1 mm in the mesio-distal direction in three animals. On the second day the gingiva acquired a cyanotic-pink color, when traumatized with a needle it was cyanotic at the injection site (Figure 1), the gingival sulcus was not probed. On the fourth day the consistency of the gingiva became friable, the lower incisors had mobility of more than 1 mm in the mesio-distal direction. By the 7th day the symptoms of inflammation increased, periodontal pockets up to 4 mm deep were probed in 9 animals, periodontal pockets 4–5 mm deep were probed in 1 animal. A more objective method of assessing tooth mobility, using Periotest S device, did not give results due to the discrepancy between the surface area of the crown part of the rat tooth and the surface area of the working part of the device striker. The complex index of inflammation in the main group for the number of animals before the experiment was equal to 4, after the experiment (on the 7th day)—55; for the number of DAS: before the experiment—9, after the experiment—195. The complex index of inflammation in the control group for the number of animals before the experiment was equal to 5, after the experiment—6; for the number of DAS: before the experiment—9, after the experiment—11. Differences in the results of examination of animals before and after the experiment in the control group were insignificant (*p* > 0.05), and in the main group the statistical significance was at the level of *p* ≤ 0.05. The differences in the examination results between the animals of the main and control groups after the experiment were statistically significant (*p* ≤ 0.05).

### 3.2. Histologic Study

The results of the histologic study are of particular interest. To evaluate the histological features of the periodontal tissues, sagittal slices were made along the longitudinal axis of the crown parts of the central incisors of rats. In all samples obtained from rats of the control group, normal alveolar bone, and marginal epithelium, as well as the absence of inflammation were observed. In the main group on the 7th day of the experiment a picture of developed inflammation was observed (Figure 2). The slice samples showed signs of exudative inflammation—leukocytic infiltration, macrophagal reaction in the intrinsic lamina of the mucosa and in the epithelium, epithelium stratification at the level of the layer of “spiny” cells (Figure 3). The ridges of the basal cell layer are smoothed, which is associated with changes in connective tissue fibers. Multilayer squamous keratinizing sulcular epithelium was unevenly thinned, detached from the epithelium of attachment. Serous, purulent, and hemorrhagic (erythrocytic and leukocytic infiltration) inflammation in the intrinsic lamina of the mucosa was observed in some samples (Figure 4). Expansion of granulation tissue with inflammatory infiltration in the gingival tissue (Figure 5), fibrinoid swelling of the gingival connective tissue, obstruction-free collagen fibers were determined in the obtained samples (Figure 6). Changes in the bony structures of the jaws also indicate pronounced periodontitis. In alveolar processes irreversible osteoclastic resorption of bone structures: inflammatory infiltrate in the apex of the alveolar process, destruction of bone tissue on the periosteum and endosteum sides, non-cellular bone resorption on the endosteum side—destruction of bone structures because of autolysis (Figure 7).

## 4. Discussion

Rats are widely used for experimental modeling of periodontal diseases and have several advantages: small size, low cost, known age and genetic background, controlled microbiota, ease of care and handling; they are characterized by the development of highly reproducible inflammatory process in the periodontium. The microbiota of the periodontal pocket of the rat in purulent-inflammatory processes includes Gram-negative bacilli, representatives of the genera *Streptococcus*, *Prevotella*, *Veilonella* and *Aggregatibacter* [38]. There are histological peculiarities in the structure of gingival tissues: the epithelium of the gingival sulcus in rodents is orogenic. In addition, there are a few physiological features of the rat dentoalveolar system: rat teeth grow constantly, and the intensity of blood supply is higher than in humans.

In known models of experimental periodontitis in laboratory animals, as a rule, oral gastric tube, periodontal inoculation, and ligature are used, besides, models with application of acute mechanical trauma are known [9]. In particular, the method of obtaining a model of chronic periodontitis in rats is known [16], which is carried out under anesthesia by applying acute mechanical trauma to gingival tissues. For this purpose, under general anesthesia on the background of thiopental anesthesia, a 12 mm long all-metal needle is inserted into the gingival margin from the vestibular surface of the right lower tooth from the medial edge closely along the tooth with a small force for tissue rupture and introduction into the bone tissue of the jaw. The disadvantage of the method is significant traumatization of tissues and stress for the animal, which can be the cause of oxidative stress and tissue hypoxia, negatively affect the result of experimentation, leading to instability of the experimental model and its low reproducibility. In addition, the period of periodontitis reproduction is 25 days, which is quite long. Needle insertion is performed under general anesthesia against the background of thiopental anesthesia, which is difficult to reproduce, given that sodium thiopental belongs to the list of potent substances, and its use is limited in the Russian Federation.

The oral probe model involves the administration of human bacterial strains via an oro-esophageal probe or micropipette. This experiment lasts from 4 to 8 weeks and is characterized by low replicability: the magnitude of bone mass loss is not always reproducible, and the systemic nature of the infection leads to inconsistent results [9].

The periodontal implantation model involves local microinjection of bacteria or lipopolysaccharide of bacterial origin, directly into the gingival papillae between the first, second, and third molars of the maxilla. The time frame for reproduction of the model is 20 to 30 days and considers only the infectious nature of periodontitis [9].

Given the well-known role of the host inflammatory response in the pathogenesis of periodontitis, there are experimental models suggesting the induction of inflammatory disease through the development of spontaneous periodontitis by modifying the diet for 16–24 weeks, which leads to the formation of microbial biofilm in the oral cavity [10,39,40].

An extremely interesting and promising direction, because of their potential translational applicability, is the creation of humanized mouse models. The first proposed model demonstrated that engraftment of human peripheral blood lymphocytes into immunodeficient mice allowed us to create an animal model for studying human immune reactions to periodontopathogens [9].

The most common methods of modeling periodontitis in laboratory animals include ligature application [1,8,9,10,11,12,21,22]. The method of modeling periodontitis in experiment is also known [8]. Acute periodontitis was modeled by applying under thiopental anesthesia at a dose of 10 mg/kg, a metal ligature at the gingival margin of the lower incisors in an eight-shaped manner, which is fixed to the crest of the alveolar process with a silk thread. From the second to the seventh day, daily, the rats are placed in cages of 0.008 m^2^ for six hours, and from the 1st to the 7th day the rats are on a high-carbohydrate diet according to Evdokimov without water restriction. Periodontitis develops on the 7th day. The disadvantage of the method is that the local effect is limited to the use of metal ligature only and does not consider the infectious nature of periodontitis. In addition, this model does not consider the pathogenetic mechanisms of periodontitis development: microcirculation disorders (ischemia, stasis, increased vascular permeability), antioxidant defense disorders and others. In addition, the model assumes obviously stressful conditions (housing) for animals.

However, there are significant limitations that apply to ligature-induced methods: it is difficult to establish this model in mice and rats because of the narrower mouth space, traditional models tend to be reproduced after 21 days and cannot reflect the specific phase of periodontitis. The methods presented above do not investigate whether ligature-induced bacteria match human bacteria. The authors state that further study of the microbiota composition and mechanisms of pathogenesis will make this modified model more convincing [11].

For the first time we have theoretically justified and tested a new experimental model of periodontitis in rats. The positive effect of the presented technique in comparison with the existing methods consists in the combined use of various local factors aimed at disturbance of functional integrity of periodontal tissues, strengthening of periodontal pathogenic microbiota activity, disturbance of microcirculation in the gingiva, which agrees with the modern theory of periodontitis occurrence and pathogenesis as a multifactorial disease; periodontitis is modeled in short terms (up to 7 days). Each year, more data on the relationship between periodontal disease and systemic diseases are emerging. However, the underlying mechanism for each disease is different and this makes its reproducibility in laboratory animals challenging, due to the different homeostasis in humans and animals. In this case, model periodontitis can be a basis for studying the relationship between periodontitis and systemic diseases [20]. Our model allows us to analyze most aspects of periodontal diseases, including bacterial interactions and dysbiosis, inflammatory reactions of the periodontium, and bone biology.

Experimental models of periodontitis in laboratory animals are promising since, at least in part, they can reproduce clinical, molecular, and histologic features of human periodontitis [9,41,42].

However, there are still several limitations to the use of our proposed model. One of the main ones is the formation of a clinically significant periodontal pocket at the site of induced inflammation. We failed to obtain a clinically significant periodontal pocket during the experiment (in more than one case). We assume that this is due to the peculiarities of blood supply to the rat head and high speed of reparative processes. It is difficult to study the effectiveness of new methods of periodontitis treatment by irrigation and instillation of periodontal pockets with antimicrobial and anti-inflammatory agents on our model because of the insignificant (up to 4 mm) depth of the periodontal pocket. The small size of the dentoalveolar segment also limits the use of classical methods of treatment of periodontitis, including the use of instruments for mechanical treatment of the root surface.

## 5. Conclusions

The new model of periodontitis in rats allows us to consider the reaction of animal tissues at the histological and macroscopic level before exposure and at the peak of developed inflammation.The periodontitis model leads to intensive infiltration of rat periodontal tissues with inflammatory cells.The developed technique allows modeling periodontitis leading to disruption of epithelial and connective tissue attachment integrity, destruction of alveolar bone.The new model of periodontitis is reproduced in a short time (up to 7 days).The methodology of periodontitis modeling does not include the use of traumatic elements for animals and obviously stressful conditions of animal housing.

## Figures and Tables

**Figure 1 dentistry-11-00219-f001:**
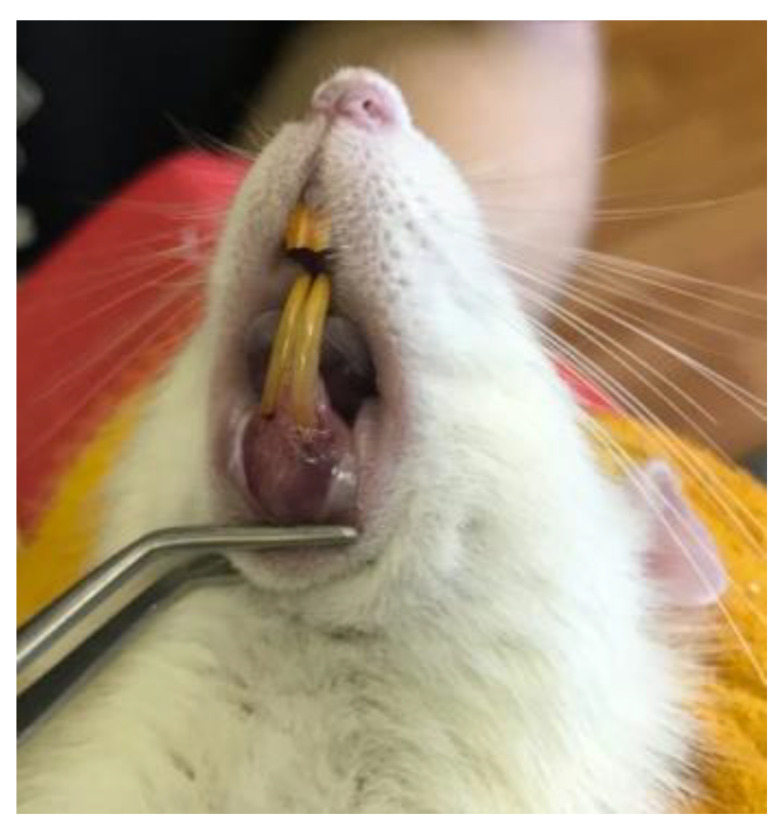
Cyanotic appearance of the gingival mucosa in the low incisor region of a rat after the injection was performed.

**Figure 2 dentistry-11-00219-f002:**
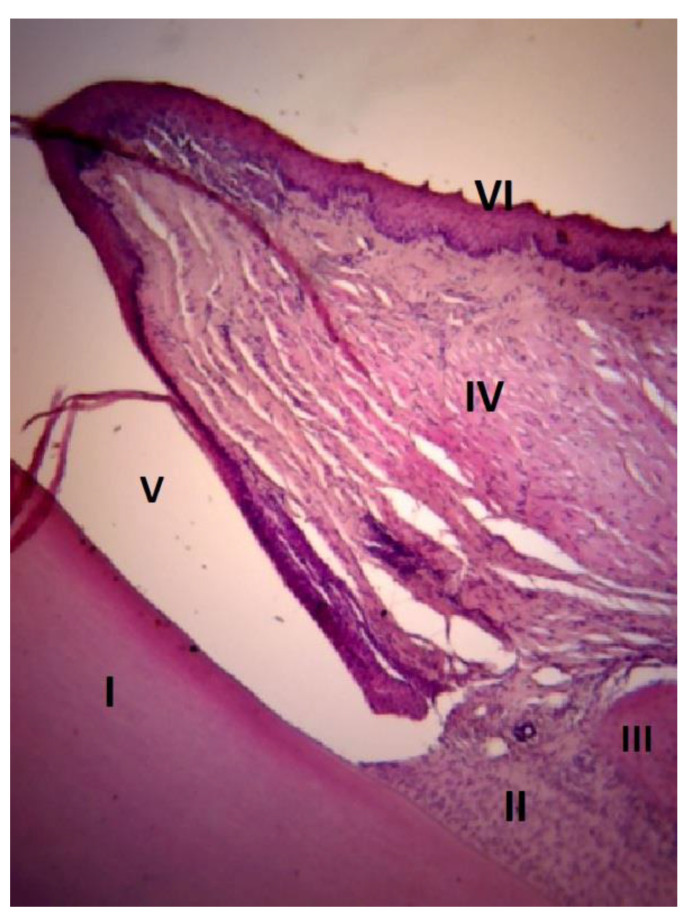
Inflammation in rat periodontal tissues, hematoxylin and eosin (magnification ×10): I—dentin of the tooth root; II—connective epithelium; III—dental alveolus with detached mucous laminae; IV—fibrinoid swelling of gingival fibers; V—detachment and irregular thinning of sulcular epithelium; VI—smoothed ridges of the basal layer of the gingival epithelium.

**Figure 3 dentistry-11-00219-f003:**
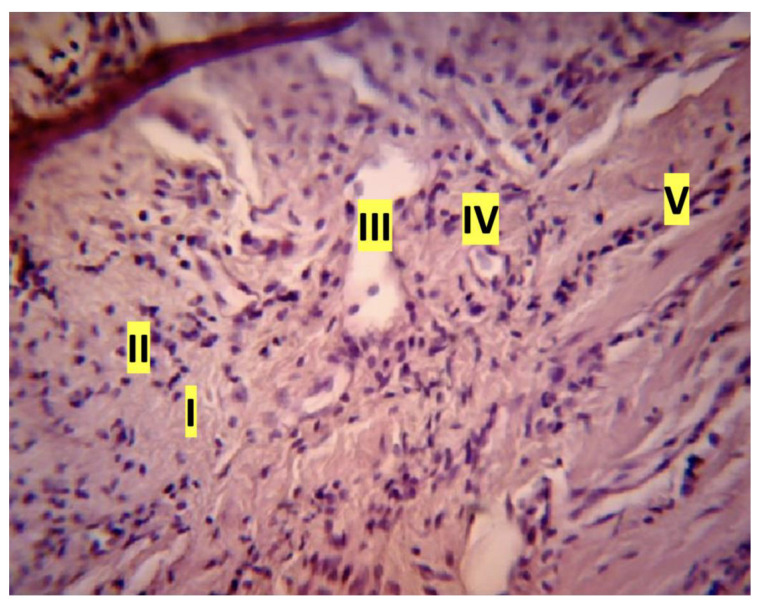
Exudative inflammation of rat gingival tissue: macrophage reaction in the mucosal lamina, hematoxylin and eosin (magnification ×100): I—edema-swelling, disorientation of collagen fibers; II—the beginning of leukocyte infiltration of the tissue; III—dilatation of the venule with parietal standing of macrophages; IV—macrophage in the cross section of the capillary; V—accumulation of leukocytes in the capillaries (longitudinal section of the vessel)).

**Figure 4 dentistry-11-00219-f004:**
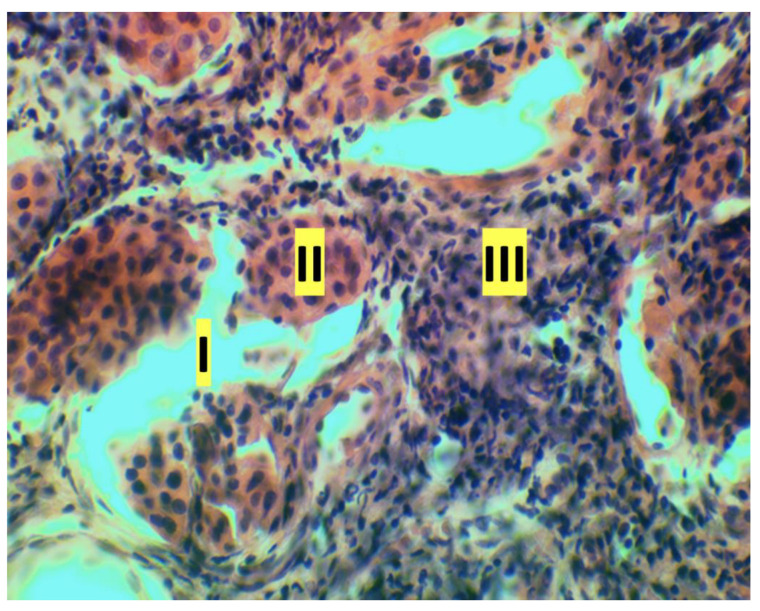
Purulent inflammation of rat gingival tissue, hematoxylin and eosin (magnification ×40): I—drain foci of purulent exudate; II—erythrocyte infiltration: translucent erythrocytes (red) are visible on the section, delimited by a leukocyte shaft; III—leukocyte infiltration.

**Figure 5 dentistry-11-00219-f005:**
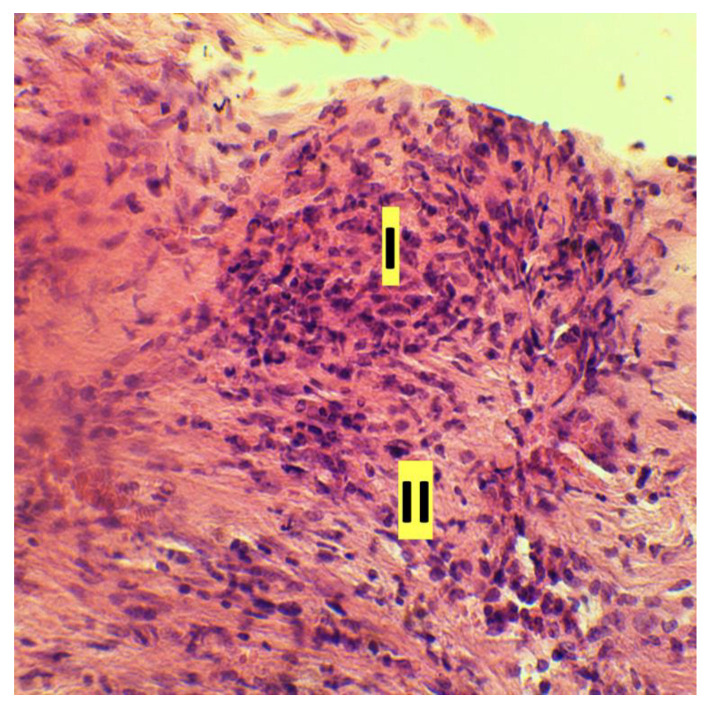
Granulation tissue overgrowth with inflammatory infiltration in rat gingival tissue, hematoxylin and eosin (magnification ×40): I—pronounced proliferation of fibroblasts; II—signs of collagenogenesis.

**Figure 6 dentistry-11-00219-f006:**
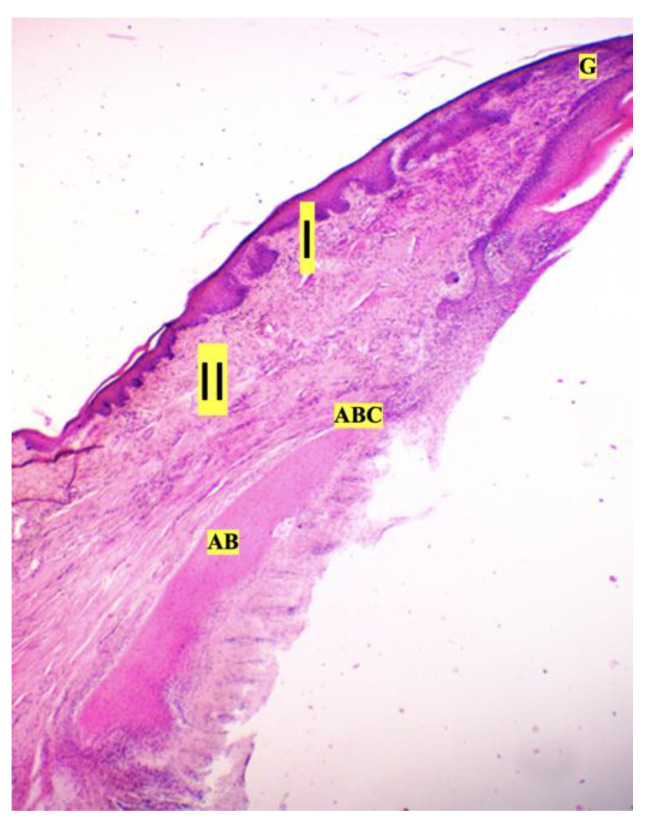
Tectorial epithelium of rat gingiva, hematoxylin and eosin (magnification ×10): AB—alveolar bone; ABC—alveolar bone crest; G—gingiva; I—fibrinoid swelling of the gingival connective tissue; II—obstruction-free collagen fibers.

**Figure 7 dentistry-11-00219-f007:**
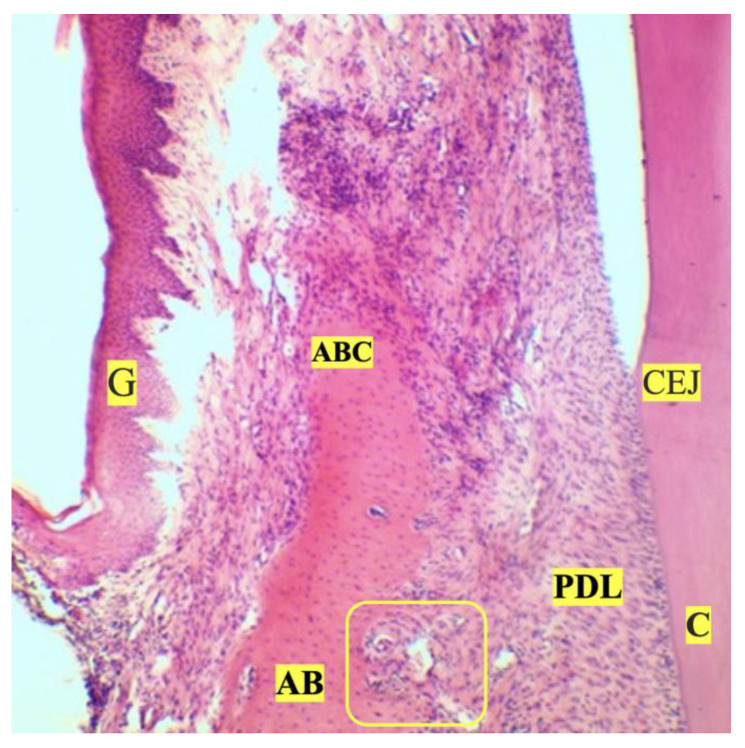
Irreversible osteoclastic resorption of rat alveolar bone structures, hematoxylin and eosin (×10 magnification): AB—alveolar bone; ABC—alveolar bone crest; G—gingiva; CEJ—cementoenamel junction; yellow border—irreversible osteoclastic resorption; PDL—periodontal ligament.

**Table 1 dentistry-11-00219-t001:** Results of examination of animals before and after simulation of experimental periodontitis.

Characteristics	Control Group				Main Group			
	Number of Animals		Number of Dentoalveolar Segments		Number of Animals		Number of Dentoalveolar Segments	
	Before	FOR 7 Days	Before	For 7 Days	Before	For 7 Days	Before	For 7 Days
Gingival hyperemia	0	1	0	2	0	9	0	36
Gingival cyanosis	2	2	4	4	2	2	5	8
Gingival bleeding	0	1	0	0	0	10	0	40
Gingival is loose, edematous	0	1	0	0	0	11	0	44
Tooth mobility, up to 1 mm in the mesiodistal direction	3	3	5	5	2	3	4	2
Mobility of teeth, more than 1 mm in the mesiodistal direction	0	0	0	0	0	10	0	40
Probing with a depth of 3–4 mm	0	0	0	0	0	9	0	23
Probing with a depth of more than 4 mm	0	0	0	0	0	1	0	2
Complex index of inflammation	5	6	9	11	4	55	9	195
*p*-value (Wilcoxon criterion)	*p* > 0.05		*p* > 0.05		*p* ≤ 0.05		*p* ≤ 0.05	

## Data Availability

The data presented in this study are available on request from the corresponding author.

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
