# Peer review of "A New Way to Model Periodontitis in Laboratory Animals"

_dentistry, 2023, doi:10.3390/dj11090219_

Round 1

Reviewer 1 Report

Comments about: A new way to model periodontitis in laboratory animals: 

1. Why do you use wire? Can it cause pain? Was it possible to control the variable pain?

2. Rats have a constant rash of the anterior incisors. How can you control that?

3. The plaque dental of patients placed under the wire: Was the dental plaque of a single patient used in all rats? Or do you use a patient's dental plaque for each rat? If it were the dental plaque of a patient for each rat, the variability of the results can be very wide.

4. The conditions with nicotine and ethyl alcohol had no controls, I mean that only 2 groups (main and control) do not deliver baseline information. For example, there is no group without ligation. I suggest having controls for each group: First group: control without ligature. Second group: only ligature. Third group: ligature with nicotine. Fourth group: ligature with ethyl alcohol. Fifth group: ligature with nicotine and ethyl alcohol.

5. Will the use of periotest S be appropriate? How can you know the depth at the probe of a periodontal pocket under health conditions in a rat?

6. The authors didn’t show a histologic study of the control group.

7. In the figure 3, 4, 5, 6 and 7 no reference is made to what is mentioned in the figure. 

8. I suggest to use once magnification. Or use two, but with zoom. For example: x40 and x 100. Because, when you use a magnification x10 didn’t see the figure

9. I suggest to use microCT to evaluate the bone and the resorption that occurs in the different conditions. 

10. I suggest to review: Optimization of the ligature-induced periodontitis model in mice. J immunol methods 2013. Is a old paper but is the basis about a induced periodontitis in animals.

Author Response

Dear reviewer!

Thank you for the time you have devoted to our work. Your comments were useful to us and greatly contributed to the improvement of our work.

We will be consistent in answering your questions.

  1. Why do you use wire? Can it cause pain? Was it possible to control the variable pain?

Thanks for the question.

The wire can cause pain, but this model is standard for the formation of periodontitis. At the same time, pain cannot be controlled - animals do not have abstract thinking and, accordingly, there is no evaluative attitude towards pain.

  1. Rats have a constant rash of the anterior incisors. How can you control that?

Thanks for the question.

The resulting model maintained a persistent periodontal lesion, which did not require control of tooth growth.

  1. The plaque dental of patients placed under the wire: Was the dental plaque of a single patient used in all rats? Or do you use a patient's dental plaque for each rat? If it were the dental plaque of a patient for each rat, the variability of the results can be very wide.

Thanks for the question.

On lines 136-137 we describe a technique for collecting plaque obtained from one patient. Dental plaque from a patient with severe chronic periodontitis, collected by scraping from the crown and cervical areas of the teeth, transported for no more than 2 hours in a sterile container, was placed under the wire using dental a spatula

  1. The conditions with nicotine and ethyl alcohol had no controls, I mean that only 2 groups (main and control) do not deliver baseline information. For example, there is no group without ligation. I suggest having controls for each group: First group: control without ligature. Second group: only ligature. Third group: ligature with nicotine. Fourth group: ligature with ethyl alcohol. Fifth group: ligature with nicotine and ethyl alcohol.

Thanks for the question.

The comparison was and can be found in table 1 (line 215)

  1. Will the use of periotest S be appropriate? How can you know the depth at the probe of a periodontal pocket under health conditions in a rat?

Thanks for the question.

Yes, in the case of an experiment, a hardware estimate is permissible in terms of absolute units to eliminate operator error.

In the case of a healthy rat, there is no need to talk about the presence of a pathological pocket, because no dentoalveolar lesion

  1. The authors didn't show a histologic study of the control group.

Thanks for the question.

We did not need to show a healthy periodontium (line 144-145) because the animals of the control group did not receive any experimental influence on periodontal tissues.

  1. In the figure 3, 4, 5, 6 and 7 no reference is made to what is mentioned in the figure.

Thanks for the question.

Damaged areas are objectively visible on histological preparations, so we did not overload the image with redundant pointers

  1. I suggest to use once magnification. Or use two, but with zoom. For example: x40 and x 100. Because, when you use a magnification x10 didn't see the figure

Thanks for the question.

We chose an increase at which changes are clearly visible depending on pathological changes in tissues. Therefore, we consider the x100 magnification for all samples not valid

  1. I suggest to use microCT to evaluate the bone and the resorption that occurs in the different conditions.

This was not part of the aims and objectives of our study. Thank you, we will take it into account in future work

  1. I suggest to review: Optimization of the ligature-induced periodontitis model in mice. J immunol methods 2013. Is a old paper but is the basis about a induced periodontitis in animals.

Thank you, we agree, we will add – this research into 38 of the list of references

Reviewer 2 Report

In this study, the authors propose a new method of accelerated modeling of experimental periodontitis on laboratory animals (white rats). The study is well conducted, and the results are adequately illustrated and interpreted. Even if there is not a great similarity between human and rat oral microbiota, the study is promising, because it provides important preliminary data before the application of similar protocols on humans.

In the discussion section, the differences between the oral microbiota and that of mice should be presented more exhaustive. It would probably have been useful to present more details related to dental plaque from the patient with chronic periodontitis.

I think the conclusions section should be written in text form. As it is written now, it has rather the format of key points.

Overall, the article is publishable, although it could be slightly improved according to the recommendations.

Best wishes,

Assoc. Prof. Dr. Stana Paunica

Author Response

Dear reviewer!

Thank you for the time you have devoted to our work. Your comments were useful to us and greatly contributed to the improvement of our work.

We will be consistent in answering your questions.

It would probably have been useful to present more details related to dental plaque from the patient with chronic periodontitis.

Thank you for your question. Our study was devoted to an experiment on an animal and people (more precisely, 1 patient) did not participate in it, except for plaque donation.

  1. I think the conclusions section should be written in text form. As it is written now, it has rather the format of key points.

Thank you, but we believe that in this form the conclusions are more concise and understandable, so we will leave it unchanged

Reviewer 3 Report

I have read the document up to the materials and methods section. As I observed many limitations and points to be clarified, I interrupted my review because, in my opinion, the objective and the materials and methods used have too many limitations to consider a publication of this work.

In the introduction:

1-    Periodontitis is a chronic inflammatory disease triggered by oral dysbiosis. It is this inflammation that leads to tissue destruction, and it is this inflammation that is also important to model > the inflammatory component of the disease is not sufficiently highlighted in the introduction.

2-    I don't understand why the authors introduce the concept of endo perio lesions, as this is not an objective of the model they are developing, and these lesions have their own specific characteristics. For me, it's irrelevant.

3-    certain notions are wrong or poorly explained (or formulated). Here are a few examples: "creating an artificial dental plaque using dental cement"; "bone pocket”; "epithelial connective tissue".

4-    The most important point is that the introduction does not present the reasons why a new periodontitis model is needed.

Therefore, in the manuscript, authors should provide an overview of the models already developed, with their characteristics and limitations. A model is defined by the way periodontitis is developed (ligature, dysbiosis...), the site where it is induced when ligatures are useful (mainly molars, as these teeth are not continuously growing), or by the animal used (and in particular its strain, as certain environments are favorable to periodontal destruction). We can also add the nature of the bacteria used. Moreover, the authors should provide a clear analysis of periodontal destruction (based on statistical analysis of connective tissue loss, bone loss and periodontal pocket formation between groups of animals ) before attempting to model the role of environmental factors such as nicotine consumption in this model.

Therefore, in this work, authors should consider the following points (not exhaustive):

- why do authors used only female ? Many authors have used only male mice to rule out the role of estrogen in the study of bone resorption in periodontology. Nowadays, the importance of studying both males and females is recognized, in order to obtain scientific results that affect both men and women, who may have specificities for certain diseases.

- The fact that incisors are more accessible is not a good explanation from a scientific point of view (unlike molars, these teeth grow continuously, which obviously has an impact on periodontal physiology). .....

- Why do they use a high-carbohydrate diet (this type of diet has been used in the past in a hamster model of periodontitis and was sufficient to trigger periodontitis due to the adherence of food and, consequently, plaque to the teeth)? Here, the authors have used a ligature around the teeth and in this type of model, in my opinion, such a regime is not necessary. Do the authors want to mimic certain environmental factors as they do with nicotine injection (....). It's not clear)?

- The authors used a patient's dental plaque: what is the composition of this plaque? How was it collected and preserved prior to application? Does the ethical agreement cover this point? How can the model be reproduced if a patient's dental plaque is used without having been defined beforehand (when the authors reproduce their experiment, which dental plaque will they use?)? ? what are the patient's characteristics?....

- The statistical justification of the number of animals used is not clear for me.

In the bibliography:

1-    Many references are inappropriate. Authors need to find and analyze appropriate source articles. For example, the use of the reference by Marchesan J et al. for epidemiological questions in the introduction is inappropriate.

2-    One third of the references are published in Russian (I haven't checked all these references, but they are indicated "(in Russ.)"> authors should use references that can be read by everyone, and in fact published in English, which remains the international language used in the scientific world.

3-    Some references are missing. (e.g. in the Introduction : “all these methods have disadvantages…. and difficulty to reproduction”.)

Author Response

Dear reviewer!

Thank you for the time you have devoted to our work. Your comments were useful to us and greatly contributed to the improvement of our work.

In the introduction, the information described on lines 81-88 is devoted to the modeling of periodontitis.

We have also added information on pathogens that cause severe forms of periodontitis in humans (lines 53-59).

In the experiment itself, the morphological characteristics of the biofilm were not in the focus of attention, because It was important for us to get periodontitis. Donor plaque was necessary to trigger periodontal tissue damage as a model and did not imply etiopathogenetic treatment, and therefore a detailed analysis of the microflora was not carried out. The materials and methods (line 135-137) indicate that the patient had severe periodontitis. We draw your attention to the severe course of periodontitis in a donor patient

We will be consistent in answering your questions.

  1. Why do authors use only female? Many authors have used only male mice to rule out the role of estrogen in the study of bone resorption in periodontology. Nowadays, the importance of studying both males and females is recognized, in order to obtain scientific results that affect both men and women, who may have specificities for certain diseases.

Thanks for the question.

If we take initially healthy rats, then the structure of the bone tissue corresponds to a healthy one. Also due to the fact that the developed model of periodontitis was accompanied by pain, and the pain threshold in females is higher, which is due to nature. this was important for carrying out the experiment in time without autotrauma on the part of the animal.

  1. The fact that incisors are more accessible is not a good explanation from a scientific point of view (unlike molars, these teeth grow continuously, which obviously has an impact on periodontal physiology). .....

Thanks for the question.

In this case, the incisors were objectively better from the point of view of the experiment, because access to the molars would require an excessive opening of the mouth, which could provoke TMJ pathology and unnecessary inflammation.

  1. Why do they use a high-carbohydrate diet (this type of diet has been used in the past in a hamster model of periodontitis and was sufficient to trigger periodontitis due to the adherence of food and, consequently, plaque to the teeth)? Here, the authors have used a ligature around the teeth and in this type of model, in my opinion, such a regime is not necessary. Do the authors want to mimic certain environmental factors as they do with nicotine injection (....). It's not clear?

Thank you for this important note. This circumstance is detailed and explained during the discussion (lines 294-298)

  1. The authors used a patient's dental plaque: what is the composition of this plaque? How was it collected and preserved prior to application? Does the ethical agreement cover this point? How can the model be reproduced if a patient's dental plaque is used without having been defined beforehand (when the authors reproduce their experiment, which dental plaque will they use?)? ? what are the patient's characteristics?.…

Thanks for your question. The method of collecting and storing plaque was added to the materials and methods in lines 135-137, where the severe form of periodontitis was clarified.

  1. The statistical justification of the number of animals used is not clear for me.

According to research, it is customary to balance the need for statistical power with the requirement to minimize experiments involving healthy people, often choosing small sample sizes from 1 to 10 [28–30]. Extrapolating data on laboratory animals, we adhere to the same point of view

​​​​​​​In the bibliography:

Thanks for your guidance. We have added the following articles to the list: 6, 7, 17, 18, 30, 35, 38.

Reviewer 4 Report

Article "A New Way to Model Periodontitis in Laboratory Animals 2"

The abstract is well structured and briefly states the subject of the article.

The Keywords section contains far too many words, some unrelated to the current study.

The presentation of periodontitis models that we did not find in the article is supported. Apart from a traumatically induced periodontitis as a result of the injection and the existence of that ligature on the teeth, we have not identified any "patterns of periodontitis".

You mentioned endo-periodontal lesions, but nowhere in the study are these types of lesions described. On the histological slices, only an image of radicular dentin appears and the dental pulp is not highlighted in any stage of inflammation.

You have recorded new methods of periodontal treatment, but in the study you only make assumptions about possible future treatments. Basically, you have not induced any periodontal inflammation remission treatment.

The introduction is well written, introducing the reader to the research topic.

The material and method chapter details the study methodology.

The statistical results are not sufficient because they do not compare the control group with the main group, which I think would be more interesting for doctors.

Regarding the histological study, it is difficult to have a complete picture of the inflammatory process only on some histological images with hematoxylin and eosin. An immunohistochemical study was much more interesting.

Please specify the degree of novelty of the study.

The article contains too many self-citations by the authors.

The bibliography is quite poor.

Author Response

Dear reviewer!

Thank you for the time you have devoted to our work. Your comments were useful to us and greatly contributed to the improvement of our work.

  1. The Keywords section contains far too many words, some unrelated to the current study.

Thanks, we've fixed the keywords now look like this: animal model; experimental induced periodontitis; histological analysis; inflammation

  1. The presentation of periodontitis models that we did not find in the article is supported. Apart from a traumatically induced periodontitis as a result of the injection and the existence of that ligature on the teeth, we have not identified any "patterns of periodontitis".

Thanks for your question. This is described in lines 81-89

  1. You mentioned endo-periodontal lesions, but nowhere in the study are these types of lesions described. On the histological slices, only an image of radicular dentin appears and the dental pulp is not highlighted in any stage of inflammation.

Thank you for the comment, we agree with it and have edited the document to remove the information about the endo-perio of the lesion

  1. You have recorded new methods of periodontal treatment, but in the study you only make assumptions about possible future treatments. Basically, you have not induced any periodontal inflammation remission treatment.

Thank you for your comment, we agree and corrected the text by removing this information.

  1. The statistical results are not sufficient because they do not compare the control group with the main group, which I think would be more interesting for doctors.

Thanks for your question. This is described in table 1 (line 214)

  1. Regarding the histological study, it is difficult to have a complete picture of the inflammatory process only on some histological iages with hematoxylin and eosin. An immunohistochemical study was much more interesting.

Thanks for the comment, we will take it into account in future works, but at the development stage it was important to show qualitative changes that are objectively visible during the current histological processing.

  1. Please specify the degree of novelty of the study.

The data obtained show that the developed model of periodontitis is highly effective and illustrative. At the same time, the animal does not experience systemic lesions that can adversely affect the course of the experiment, and the model does not cause shock conditions and the death of the individual.

  1. The article contains too many self-citations by the authors.

Thank you for your comment. We took it into account and corrected the bibliography

  1. The bibliography is quite poor.

Thanks for your guidance. We have added the following articles to the list: 6, 7, 17, 18, 30, 35, 38.

Round 2

Reviewer 1 Report

Dear authors

1. While it is true that animals do not have abstract consciousness, like every animal in the world they suffer from pain, therefore, if you put a wire in their teeth it will cause them pain and therefore there could be animal abuse. Have you considered that point?

2. Whether or not there is a periodontal lesion, the anterior incisors of rodents have a constant eruption. That means that the record they have of the periodontal sac is not accurate.

3. My question is not about the procedure, my question is about the implication of placing dental plaque on the wire and whether each rat has a patient's dental plaque placed on it. If so, each patient generates a different microbiota and also, what are the clinical parameters that are detected in the patient.

4. My question is directed to the fact that not only 2 groups can provide data regarding nicotine and ethyl alcohol conditions. Each group to be worked on should have controls.

5. How big is a periodontal pocket in a rat and how big is a gingival sulcus in a rat? Is there any paper that proves this?

6. It is important to have and show control groups as this way you can see the comparison between the experimental group and the control group. For example, higher numbers of inflammatory cells in disease conditions.

7. Figures should always point out what you want to show, even if it is obvious.

8. Unfortunately, if several magnifications are placed on the figures, it is not possible to observe the conditions. Therefore, even if it is obvious, it should be pointed out in the figure.

9. Unfortunately, as it is not possible to have an exact record of a periodontal pocket in a diseased condition, other parameters must be taken into account, one of them being the measurement of the alveolar bone.

Dear Editor

Author Response

Dear Reviewer!

We are grateful to you for such a detailed and careful consideration of our article, for the identified shortcomings, for your comments.

Let me give you a detailed answer to your comments.

  1. While it is true that animals do not have abstract consciousness, like every animal in the world they suffer from pain, therefore, if you put a wire in their teeth, it will cause them pain and therefore there could be animal abuse. Have you considered that point?

Lines 119-126 describe in detail the principles of working with animals, which we strictly observed during the entire experiment. We are grateful for this question, which allowed us to write in more detail one of the items in the Primary documentation, which was devoted to assessing the condition of animals during the experiment. Given the trauma in the oral cavity, we identified points of observation of the behavior of the animal and attempts to remove the ligature from the oral cavity, as well as loss of appetite, forced postures and voice accompaniment. Since the subjects in the postoperative period did not change their eating habits, did not stay in a forced position and did not try to influence the oral cavity in any way, we concluded that the developed method does not cause pain irritation.

  1. Whether or not there is a periodontal lesion, the anterior incisors of rodents have a constant eruption. That means that the record they have of the periodontal sac is not accurate.

Thanks for the question. As you know, the periodontium is not prone to regeneration, and after the destruction of the circular ligament, the periodontal lesion persists regardless of the growth of the tooth. We can add about the dynamic process in relation to the tooth, as the walls of the periodontal pocket, but not about the soft tissue basis.

  1. My question is not about the procedure, my question is about the implication of placing dental plaque on the wire and whether each rat has a patient's dental plaque placed on it. If so, each patient generates a different microbiota and also, what are the clinical parameters that are detected in the patient.

Thanks for the question. In our case, the plaque donor for experimental animals was one person, so it is legitimate to talk about one microbiota.

  1. My question is directed to the fact that not only 2 groups can provide data regarding nicotine and ethyl alcohol conditions. Each group to be worked on should have controls.

Thanks for the question. The control group itself is the control group and does not need additional supervision. If you mean placebo control, then this was not included in the experiment, because. does not imply damage.

  1. How big is a periodontal pocket in a rat and how big is a gingival sulcus in a rat? Is there any paper that proves this?

Thanks for the question. The depth of the periodontal pocket is shown in Table 1.

As for the depth of the sulcus, it was not determined in clinically significant terms. This is supported by a study by Dharmawati, I. A. et al (reference number 42)

  1. It is important to have and show control groups as this way you can see the comparison between the experimental group and the control group. For example, higher numbers of inflammatory cells in disease conditions.

Thanks for the question. Cell count was not included in the goals and objectives of the study.

  1. Figures should always point out what you want to show, even if it is obvious.
  2. Unfortunately, if several magnifications are placed on the figures, it is not possible to observe the conditions. Therefore, even if it is obvious, it should be pointed out in the figure.

Thank you for your comments, we have added clarifying captions and reworked the images.

  1. Unfortunately, as it is not possible to have an exact record of a periodontal pocket in a diseased condition, other parameters must be considered, one of them being the measurement of the alveolar bone.

In an acute inflammatory reaction of the periodontium, local activation of phagocytes leads to a surge of oxidative reactions and the release of a huge number of hydrolytic enzymes, which destroy microbial cells and primarily contribute to the leaching of the mineral components of the bone while preserving the organic matrix. With the rapid development of inflammation under the influence of periodontal pathogenic flora, degradation of the extracellular matrix occurs, which is not immediately reflected in the alveolar bone. Because the duration of the experiment was 7 days, so it was not advisable to measure the alveolar bone.

Reviewer 4 Report

Dear authors

The changes you made are good.

Author Response

Dear reviewer!

Thank you for your appreciation of our work and for the time you spent on reviewing